# Design of a Real-Time Gas-Exchange Measurement System for Crop Stands in Environmental Scenarios

**Hans-Peter Kläring and Oliver Körner \***

Leibniz-Institute of Vegetable and Ornamental Crops (IGZ), Theodor-Echtermeyer-Weg 1, 14979 Großbeeren, Germany; klaering@igzev.de

**\*** Correspondence: koerner@igzev.de

**Abstract:** In contrast to conducting measurements on single plants, canopy gas exchange monitored continuously and for large batches of plants can give high-value data for crop physiological models. To this end, a system including eight airtight greenhouse cabins with a ground area of 28.8 m$^2$ and a volume of 107.8 m$^3$ each was designed for measuring the $CO_2$ and $H_2O$ gas exchange of crop stands following the general principle of semi-open chambers. The measuring facility consists of a set of mass flow meters allowing air exchange rates between 0.5 h$^{-1}$ and 19 h$^{-1}$ (i.e., m$^3$ gas per m$^3$ greenhouse air per hour) and $CO_2$ supply rates up to 4 L min$^{-1}$ (i.e., ca. 14.9 g m$^{-2}$ greenhouse h$^{-1}$) and sensors for measuring the concentrations of $CO_2$ and $H_2O$. There are four separated belowground troughs per cabin for the root environment that can be operated as individual gas exchange chambers measuring the belowground gas exchange for example root zone respiration. This paper outlines a demonstration of the possibilities and constraints for measuring crop gas exchange in combination with crop model validation for larger crop stands under various conditions and discusses them along with examples.

**Keywords:** crop photosynthesis; greenhouse gas exchange; crop growth models; photosynthesis; evapotranspiration; greenhouse physics; greenhouse climate control

## 1. Introduction

In many ecological and agricultural studies, the instantaneous responses of plants to environmental effects is of fundamental interest, and non-invasive online measurements of carbon dioxide ($CO_2$) and water ($H_2O$) gas exchange of plants and canopies are excellent techniques to measure such responses. Several types of equipment and systems have been developed to meet this need. $CO_2$ and $H_2O$ gas exchange techniques with handhold infrared gas analyzers (IRGA) or other technology is common practice in crop physiological research [1]. The advantage of this type of gas-exchange chambers on a laboratory scale is that they generally allow accurate control of the environmental conditions. The size of these chambers, however, is usually relatively small, often measuring only a few cm$^2$ of a single leaf. Interpretation of data and theoretical upscaling is difficult due to the large variation within an individual leaf as well as between leaves in a plant or crop. The interest in upscaling physiological processes in space and time has been grown since the 1980s and new measurement techniques such as eddy-covariance or remote sensing procedures have been developed [2].

While these techniques are commonly used to quantify $CO_2$ exchange on the ecosystem scale [3], their disadvantage is the large ecosystem scale and the relatively long time frame of 24 h. In protected cultivation in controlled environments, e.g., in greenhouses, processes with a long time frame within days are next to fast time responses within seconds or minutes of high importance for optimized climate control strategies [4,5]. To create data for model calibration and validation on a greenhouse scale and in controlled environments, for optimized climate control, and for model-based decision

support systems, larger systems measuring single plants [6], or a group of small plants on laboratory scale [7], or potted plant trays [8,9] have therefore been developed. Although experimental data from those systems were valuable for use in protected cultivation, none of them were capable of separately measuring shoot and root zone gas exchange, an often-underestimated factor in crop photosynthesis models. An attempt to separate measurements of shoot and root has therefore been made [10,11] including carbon isotope discrimination [11].

Next to measuring systems for complete plants, even larger systems for crop photosynthesis measurement have been designed since the 1990s as summarized in Table 1. Most of the gas-exchange systems described above were developed and used for specific research topics, mostly for photosynthesis model calibration or validation. Some systems were used repeatedly to measure photosynthesis of different crops under various environmental conditions [12–15]. For other systems, accurate operational reliability was proven, and, as an example, crop photosynthesis was measured for fitting light response curves [16,17].

**Table 1.** Timely development of experimental whole crop gas-exchange greenhouse measurement systems.

| Year | Ref. | Method | Gas Supply | Type | Loc. | Size, $m^2$ | Crops |
|------|------|--------|-----------|------|------|---------|-------|
| 1992 | [16] | Null-balance at ambient $CO_2$ with proportional pure $CO_2$ injection | Technically pure $CO_2$ with mass flow controller | open | UK | $1 \times 162$ | Cucumber |
| 1994 | [12] | Air exchange rates measured online using nitrous oxide ($N_2O$) as tracer gas. | Technically pure $CO_2$ with mass flow controller | semi-open | NL | $4 \times 192$ | Cucumber Tomato Sweet pepper |
| 2007 | [13] | Proportional pure $CO_2$ injection with max. $10.2$ g m$^{-2}$ h$^{-1}$ to air conditioning unit for elevated $CO_2$ | Technically pure $CO_2$ with mass flow controller | semi-closed | NL | $2 \times 44$ | Tomato Chrysanthemum |
| 2011 | [17] | Outside air drawn through a dry cooling pad with ambient $CO_2$ | No $CO_2$ supply | open | ISR | $2 \times 360$ | Sweet pepper |
| 2015 | [14] | Supplying outside air through ducts in a range of 3 to $12$ m$^3$ m$^{-2}$ h$^{-1}$ to ambient $CO_2$ | No $CO_2$ supply | semi-open | D | $6 \times 64$ | Tomato |

In general, crop gas exchange measuring systems can be divided into systems with controllable or ambient $CO_2$ concentration, with the disadvantage of varying $CO_2$ concentration in the crop canopy, or in open or closed systems. In open systems, photosynthesis and transpiration are estimated from the differences in $CO_2$ and $H_2O$ concentration between air-intake and exhaust air. In closed systems crop gas exchange is solely measured through the amount of gas injection commonly supplied to maintain a desired gas concentration set-point.

The design of crop gas exchange facilities is always a compromise between the universality and accuracy of the respective system and the technical possibilities of a suitable solution. While small facilities (e.g., handheld IRGA or single plant chambers) were designed for repeated use in many experimental measurements, all reported large greenhouse-based gas-exchange systems were designed for certain experimental question and "disappeared" soon after.

In an endeavor to create a universal system, a facility to measure the photosynthesis of complete canopies consisting of eight cabins was designed and established at the Leibniz-Institute of Vegetable and Ornamental Crops (Großbeeren, Germany; 52°21'31.03" N, 13°18'35.78" E). The cabins can be operated in both closed-chamber and open-chamber mode. Controlled air exchange rates of up to $70$ m$^3$ m$^{-2}$ h$^{-1}$ allow operation under most environmental conditions. The system is equipped with independent root-zone gas-exchange measurement options, enabling the separation of real-time root and shoot crop gas exchange. To mimic certain atmospheric conditions for crops during $CO_2$ or $H_2O$ gas-exchange measurements (e.g., behavior in a given region or for analyzing future environmental scenarios) eight different air pollutants can be dosed into the cabins, too.

## 2. Materials and Methods

### 2.1. $CO_2$ and $H_2O$ Gas Exchange of Complete Crops

The experimental facility at Großbeeren (52°21′31.03″ N, 13°18′35.78″ E) consists of eight identical almost airtight glass-greenhouse cabins (Hans Marsmann Gewächshausbau, Münster-Wolbeck, Germany) with a ground area of 28.8 m$^2$ and a volume of 107 m$^3$ each (720 cm × 400 cm, length × width; 340 cm height at the eaves, 400 cm height at the ridge) (Figure 1). The walls are built with double thermo-glass while the roof consists of UVB-transmissible single glass. All cabins can be entered from a glass-corridor (3200 cm × 400 cm; length × width). Below the corridor, a basement with most technical equipment is located.

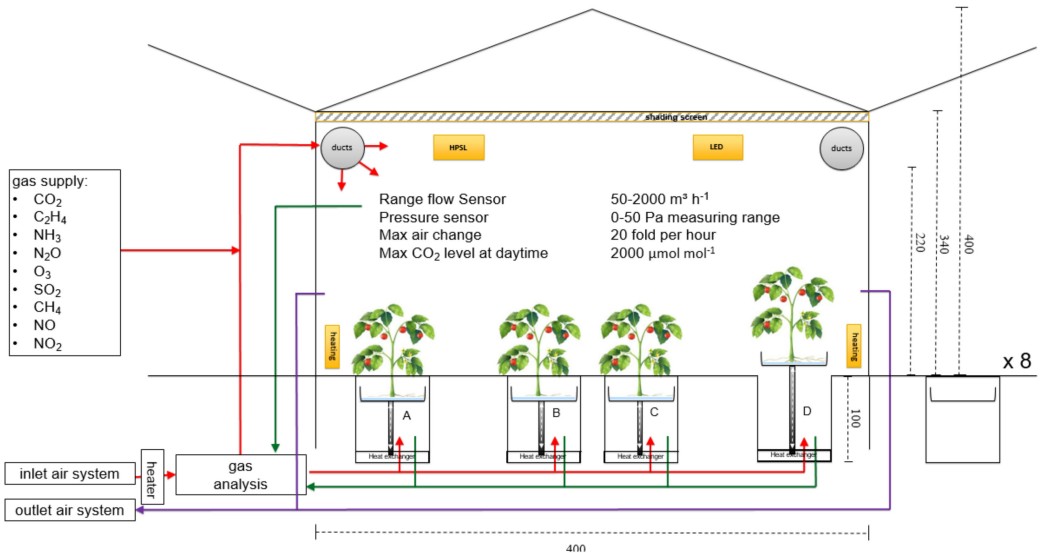

**Figure 1.** Simplified sketch of the experimental gas-exchange facility (not on scale) of one of eight cabins with four troughs (A, B, C, in separate root zone measurement mode; D in open mode for cooling the shoot environment). Red lines denote gas-supply, green lines denote gas sampling, and violet lines is the outlet air. For supplementary lighting 12 high-pressure sodium lamps (HPSL) or multi-channel LED lamps can be mounted (six lamps of each side of each greenhouse compartment). All fluxes are measured with continuous flux-meters. Irrigation and fertigation are done automatically and can be adjusted for each trough independently (not in the figure).

For the root environment, each cabin contains four troughs embedded in the floor (600 cm × 50 cm × 100 cm, length × width × depth). The troughs are movable in a vertical direction while the top is covered with white polyethylene plates when closed (Figure 1). Independent root zone temperature control is achieved by heat exchange pads in the troughs. Nutrient solution can be dosed in groups or for each trough independently.

Each of the eight cabins and each of the four troughs in each cabin can be used separately for online measurement of net $CO_2$ and $H_2O$ gas exchange. In addition, eight other environmental gases can be connected and dosed to the system in order to mimic certain environmental conditions (see Figure 1). Depending on the experimental question, it is possible to run the cabins both as open or closed chambers, while the troughs can only be operated in open chamber mode. In this mode, evapotranspiration can be measured.

### 2.2. Configuration Measuring $CO_2$ Gas Exchange

In the case of open chambers, air exchange rates can be adjusted in a range of 50 to 2000 m$^3$ air per cabin per hour; i.e., air exchange of 0.47 to 18.69 h$^{-1}$. Outside air is sucked in and supply air is pre-processed by four central ventilation units (HZG 400.4, Klimatec, Trier, Germany), each with

7.5 kW of power. The supply air can be heated when outside temperatures are low. The overpressure of the supply air can be adjusted between 0 and 5 kPa. An overpressure of 0.5 kPa allows air exchange rates up to 1200 m$^3$ h$^{-1}$. Heat from the ventilation units increases the temperature of the supply air by 2 K. The maximum air exchange rate requires an overpressure of about 1000 Pa, resulting in an air temperature increase of 4 K. The supply air is transported via stainless steel ductwork to the cabins. The air supply rate to each cabin is controlled by one 4'' valve and two 6'' valves (R411 and R439, Belimo, Stuttgart, Germany) and measured using a high-precision ultrasonic flow meter (Digital Flow XGM8681, GE Sensing and Inspection Technologies, Hürth, Germany). In order to achieve high accuracy, low air supply rates are controlled by the smaller valve only, while the larger valves become incrementally more involved as air supply rates increase. As a result, the absolute measurement error is less than 2% (sensors are certified annually by the manufacturers). The supply air is distributed via perforated acrylate ducts on both sides of the cabins at a height of 2.2 m.

The cabins are almost airtight with a known minimum leakage error (used as a variable in the gas-exchange equation). The supplied air is actively removed by four exhaust air units (HZG 063.4, Klimatec; provide negative pressure between 0 and 2 kPa) via a conduit in one corner of the cabin. The suction of the exhaust air is controlled by one small and two larger dampers (ASKA110 and ASK160, Mietzsch, Dresden, Germany) with actuators (LM24A-SR, Belimo, Hinwil, Switzerland). The suction process starts using the small damper and with increasing exhaust air rates subsequently using the two larger dampers. The control of the exhaust air rate is based on a constant pressure difference between the cabin and the ambient air, measured by a differential pressure sensor adjustable up to 50 Pa (DDPT-I2-DP, Prignitz, Wittenberge, Germany). Pure technical $CO_2$ can be supplied (SIMATIC S7-300, Siemens, Munich, Germany) controlled by mass flow controllers (GF 40, Brooks Instrument, Hatfield, PA, USA). The system is capable of $CO_2$ enrichment to a limit of 3000 µmol mol$^{-1}$, while $CO_2$ absorption (scrubber) is not implemented.

An infrared gas analyzer (IRGA; URAS 26, ABB, Zurich, Switzerland) measures the concentrations of carbon dioxide ($CO_2$) and water ($H_2O$) in the cabin and the ambient air. To achieve this, ambient air and air from the cabins is continuously sucked close to the IRGA but discarded when not sampled. As such, a sample gas pump cyclically pumps ambient air and cabin air from these gas streams into the analyzer, controlled by magnetic valves (S307-05, Sirai, Bussero, Italy). The measurement signal stabilizes after 1 min, resulting in a maximum measurement cycle of 9 min i.e., when all eight cabins are included in the measurements.

Additional equipment such as an isotopic $CO_2$ cavity ring-down spectroscopy system can be fed with the URAS 26 exhaust air in order to measure the 13C/12C fraction in the outside, cabin, or trough air, respectively.

### 2.3. Calculation of the $CO_2$ and $H_2O$ Gas Exchange

The rate of net $CO_2$ gas exchange in relation to the ground area occupied by the crop ($A_{crop}$, m$^2$) herein referred to as net photosynthesis ($P_{net}$, g m$^{-2}$ h$^{-1}$) can be estimated as used in Körner et al. (2007) [13] for time steps ($\Delta t$, h) as

$$P_{net}(t) = \frac{u_s(t) \cdot \rho_s^{CO_2} \cdot (C_s(t) - C_e(t)) - \frac{1}{\Delta t} \cdot (C_e(t) - C_e(t - \Delta t)) \cdot \rho_e^{CO_2}(t) \cdot V + S^{CO_2}(t)}{A_{crop}} \tag{1}$$

where $u_s(.)$ (m$^3$ h$^{-1}$) is the supply air rate; $C_s(.)$ and $C_e(.)$ (mol mol$^{-1}$) are the $CO_2$ concentrations in the supply and exhaust air, respectively; $\rho_e^{CO_2}(.)$ (g m$^{-3}$) is the density of $CO_2$ at the temperature of the exhaust air; and $S^{CO_2}(.)$ (g h$^{-1}$) is the $CO_2$ supply rate. For simplification, it was assumed that the $CO_2$ concentration of the exhaust air would be a close approximation of the $CO_2$ concentration in the cabin, and the very marginal effect of the cabin pressure on $\rho_e^{CO_2}$ was omitted. All variables $x(.)$ are average values of a period $[t - \frac{\Delta t}{2}, t + \frac{\Delta t}{2})$. $V$ (m$^3$) denotes the cabin volume and $\rho_e^{CO_2}$ (g m$^{-3}$) indicates the density of $CO_2$ in the supply air. The supply air rate $u_s$ is provided as a standard volume flow rate,

meaning that it is related to a temperature of 0 °C and a pressure of 101,325 Pa. Therefore, must also refer to these conditions, and is a constant. The effect of evapotranspiration on the difference between supply and exhaust air rate is omitted here because it is negligible.

The rate of $H_2O$ gas exchange related to the ground area of the crop, herein referred to as evapotranspiration $E$ (g m$^{-2}$ h$^{-1}$) can be estimated analogously to $P_{net}$; however, it is necessary to take into account the possible presence of any condensation of water vapor on the cabin cover:

$$E(t) = \frac{u_s(t) \cdot \rho_s^{H_2O} \cdot (W_s(t) - W_e(t)) + \frac{1}{\Delta t} \cdot (W_e(t) - W_e(t - \Delta t)) \cdot \rho_e^{H_2O}(t) \cdot V + W_{cond}(t)}{A_{crop}} \qquad (2)$$

Here, $W_s$ and $W_e$ (mol mol$^{-1}$) denote the water vapor content of the supply and exhaust air; $\rho_s^{H_2O}$ and $\rho_e^{H_2O}$ (g m$^{-3}$) are the water vapor density at 0 °C and at the temperature of the exhaust air, respectively. The water condensed on the roof $W_{cond}$ (g h$^{-1}$) can be measured using rocker cup sensors or estimated from the measured or calculated roof temperature.

## 2.4. Removing Heat and Humidity from the Cabin Air

In the case of a low outside temperature and low crop transpiration rates, the total transpired water can be recaptured from the air by roof condensation. In this case, the air supply to the cabin can be interrupted, and the cabin can be operated in the closed chamber mode.

The troughs for the roots are equipped with mats for heating and cooling the root environment. However, when opening the troughs and lifting the tables with the plants above the level of the cabin floor (Figure 1), the trough cooling can be used for cooling the cabin air, leading to a decrease of absolute air humidity through condensation. For this to occur, the warm and humid air from the cabin roof space must be actively transported via tubes and axial fans into the cooled troughs. This expands the possibilities for measuring $P_{net}$ in closed chamber mode, while it allows lower air exchange rates at conditions with high energy load (high global solar radiation, high outside temperature) in open chamber mode.

## 2.5. Environmental Constraints for Gas Exchange Measurements

Air temperature can be adjusted by setpoints for heating and ventilation as it is common in regular climate-controlled greenhouses, as well as with controlled air supply rates at closed vents. However, roof ventilation inhibits gas exchange measurements and vent opening is to be avoided when necessary. Supplying a tracer gas such as $N_2O$ at a constant rate and estimating the air exchange rate during open roof ventilation [12,18,19] has not been tested yet.

A rough estimate of the sensitive heat transfer out of the greenhouse cabin ($q_{sensitive}$, W m$^{-2}$) per unit of cabin floor area ($A_{floor}$, m$^2$) by exchange of dry air and heat transmission through the cover under steady-state conditions can be obtained:

$$q_{sensitive} = \frac{u_s \cdot \rho_s^{air} \cdot (T_e - T_s) \cdot c^{air} + k_{roof} \cdot (T_e - T_a) \cdot A_{roof}}{A_{floor}} \qquad (3)$$

where $\rho_s^{air}$ (kg m$^{-3}$) denotes the air density at standard conditions; $c^{air}$ (J kg$^{-1}$ K$^{-1}$) is the specific heat capacity of dry air; $k_{roof}$ (W m$^{-2}$ K$^{-1}$) is the thermal transmittance of the greenhouse roof, $A_{roof}$ (m$^2$) is the cabin roof area; and $T_s$, $T_e$ and $T_a$ (°C) are the temperature of the supply, exhaust, and ambient air, respectively. $T_s$ is higher than $T_a$ due to the heat disposal of the ventilation units. For simplification, Equation (3) does not account for the effect of water vapor in the air. In addition to sensitive heat transfer, latent heat transfer ($q_{latent}$, W m$^{-2}$) due to evapotranspiration must be taken into account:

$$q_{latent} = \frac{A_{crop} \cdot E \cdot \left(\Delta H_{vap} + c^{H_2O} \cdot (100 - T_e)\right)}{A_{floor}} \qquad (4)$$

$\Delta H_{vap}$ (J g$^{-1}$) and $c^{H_2O}$ (J g$^{-1}$ K$^{-1}$) denote the vaporization enthalpy at 100 °C and the specific heat capacity of water, respectively. The heat transfer out of the greenhouse—the cooling power—must be accounted for in relation to any heat transfer into the greenhouse from global radiation passing through the greenhouse cover. In this rough calculation, effects from the side walls to the adjacent cabins, the corridor, and the outside have been omitted, as all walls are made of double thermal glass with a heat transmission coefficient of 1 W m$^{-2}$ K$^{-1}$. External screens can be used to avoid most of the solar radiation from penetrating into the cabin through the outside walls.

### 2.6. Air Exchange Rates of Closed Cabins and $CO_2$ Sources

The air exchange rates of closed cabins must be taken into account when operating in closed chamber mode. In order to measure these air exchange rates, the cabins were flushed with $CO_2$ up to concentrations of about 3000 µmol mol$^{-1}$. Next, the $CO_2$ supply was disconnected. The air exchange rate could then be obtained from the $CO_2$ depletion curve

$$\ln[C_c(t) - C_a(t)] = a_0 - a_1 \cdot t \tag{5}$$

where $t$ (h) denotes the time, $C_c(t)$ and $C_a(t)$ (mol mol$^{-1}$) denote the cabin and ambient $CO_2$ concentrations at time $t$, and $a_0$, and $a_1$ are regression parameters. Multiplying $a_1$ by the cabin volume $V$ provides the air exchange rate $u'$ (m$^3$ h$^{-1}$). Note that in contrast to the air supply rate $u_s$, the air exchange rate $u'$ is related to the density of the air in the cabin at $T_c$ (°C) during the measurement.

The quantity of $CO_2$ sources (or sinks) was estimated by adjusting a constant supply air rate in the empty cabins and calculating the $CO_2$ flux from the cabins according to Equation (1).

### 2.7. Controlling the Cabin Climate and Other Possible Treatments of the Plants

There are many possibilities to control the individual cabin environment for different aspects of crop photosynthesis and transpiration. Air temperature can be varied among the cabins by adjusting different heating set points or controlling the supply air rate. The cabins have shading screens that can reduce global solar radiation at plant level by 50%. Plant available radiation can be increased by supplementary lighting with twelve lamps of either high-pressure sodium discharge type (Master AGRO 400 W, Philips, Amsterdam, The Netherlands) with up to 210 µmol m$^{-2}$ s$^{-1}$ or with multi-spectral LED lamps with 8 free mixable and programmable light channels (LightDNA-8; Valoya, Helsinki, Finland) with a range between 380 and 780 nm with a maximum of 320 µmol m$^{-2}$ s$^{-1}$ (all channels 100%). The lamps were refurbished with an additional acryl-glass cover for water protection.

Air $CO_2$ concentration in the cabins is kept at the desired level, while relative humidity may be reduced by increasing supply air rates or increased by fogging water at a pressure of 120 bar. The latter, however, inhibits transpiration measurements.

The dosage of air pollutants, including ozone ($O_3$), sulfur dioxide ($SO_2$), nitrogen oxides (NOx, $N_2O$), methane ($CH_4$), ethylene ($C_2H_4$), and ammonia ($NH_3$) into the cabin atmosphere in order to investigate their effect on photosynthesis and transpiration. The plant itself can also be manipulated, e.g., by changing the sink/source ratio. In addition to the shoot environment, there are many options to consider the effects of treatments in the root environment on the gas exchange characteristics of the shoots.

### 2.8. Separating the Gas Exchange of the Shoot and of the Root Zone

Between 2016 and 2020 we investigated the behavior of the system for different crops as tomato, cucumber, broccoli, and apple. For all crops, the gas exchange of the aboveground plant components can be measured independently of that of the root zone. The latter has been used for measuring the effect of $N_2O$ emission in the root-zone [20]. Briefly, for root-zone $CO_2$ measurements, after closing the troughs containing the roots in an almost airtight way, ambient air is injected into the root zone

using mass flow controllers (VA 420, CS Instruments, Villingen-Schwenningen–Tannheim, Germany). The air leaves the root zone via the remaining leakages in the cover of the troughs due to the slight overpressure measured by using different pressure sensors (DDPT-I2-DP, Prignitz). At the same time, the overpressure prevents air from the cabin from penetrating into the troughs. The $CO_2$ concentration of the supply air and the air in the troughs is measured analogously to the measurements in the cabins using a second URAS 26 infrared gas analyzers. The root-zone $CO_2$ release can be calculated from the air injection rate and the difference in the $CO_2$ concentration of the air in the troughs and that of the injected air. Then, Equation (1) needs to be adjusted. At the same time, the release of other greenhouse gases such as $N_2O$ or $CH_4$ can be measured, too.

## 2.9. Crop Experiments

Three experiments with tomato crops were performed for facility testing with $CO_2$ and $H_2O$ gas exchange. In Experiment 1, tomato plants ('Komeet', De Ruiter Seeds, The Netherlands) were planted on 28 September 2015, the response of gas exchange measurements to sudden changes in photosynthetic photon flux density (PPFD, μmol [photons] $m^{-2}$ $s^{-1}$) was evaluated. In Experiment 2, tomato plants ('Komeet') were planted on 12 January 2016, the climate control options, and both the open and closed chamber mode were examined. In Experiment 3, tomato plants ('Pannovy', Novartis, UK) were sown 02 January 2018 and planted to the gas-exchange facility 22 February 2018. $CO_2$ gas-exchange of crops at different temperature and $CO_2$ levels was investigated for validation of a crop photosynthesis model.

In all three experiments (Experiment 1, Experiment 2, and Experiment 3), after developing around eight true leaves, twelve rockwool cubes with the plants were each set on a black mat of fibers on the tables, resulting in a plant density of 2 plants $m^{-2}$. Plants were supplied with a nutrient solution via drip irrigation according to a formula for Dutch horticulture [21]. A drainage solution fraction of 40% of the solution supplied was set. The tables were aligned 15 cm below the cabin floor. Next, the troughs were covered using clipboards. The shoots were fitted through holes in the covers of the troughs and trained vertically using a wire. The plants were trained weekly: all side shoots and leaves below trusses with red fruits were removed. Shoot growth was terminated when the stem reached the mounting of the wire, at a height of 2.7 m. Pollination was facilitated by vibrating flowering trusses at least ones a week.

Leaf length was measured periodically during cultivation, and leaf area was estimated using an allometric relationship [22]. The total biomass produced (harvested fruit, removed leaves, complete plants at the experiment's end) was recorded and subsamples were dried in a ventilated oven to estimate the dry matter.

## 2.10. Comparing Measured and Simulated Crop Photosynthesis

For comparison of measured crop photosynthesis in a block-shape crop (as it is the case in the current facility) with theoretical values, the light distribution within the blocked crop needs to be calculated. As such, crop stands can only be used with the comparison of measurements with the same crop architecture. For comparisons between models and measurements for model validation purposes or for parameterization, an architectural model of the block-shaped structure of the crop canopy and the photosynthesis model needs to be combined. Simple upscaling would due to the non-endless structure of the canopy yield in underestimation of measured photosynthesis. Due to that, leaf photosynthesis was scaled up to the crop level with a geometric radiation absorption model as used by Körner (2003) [23]. Photosynthesis of separate leaves was determined with photochemical efficiency ($a_l$) and maximum leaf gross photosynthesis ($P_{gl,max}$) of the negative exponential light response curve [24]. $P_{gl,max}$ and $a_l$ were calculated with several biochemical key processes from the FvCB-Model for leaf $CO_2$ assimilation [25] that have been summarized in a simplified description [26,27]. A homogeneous distribution of $CO_2$, temperature, and humidity inside the canopy was assumed and morphological

differences between leaves of different ages and locations were neglected. Only the vertical difference of stomata resistances was accounted in the vertical calculation of $P_{gl,max}$ and $a_l$.

A model was developed that calculated diffuse and direct photosynthetic active radiation (PAR) absorbed by individual leaves inside the block-shaped plant stand. A rectangular 3-dimensional grid of 4.000 points ($P_{xyz}$; 20 points along the width, 20 points along the depth, and 10 points along the height of the block) was adopted. Diffuse and direct PAR were first determined outside the greenhouse and transmission was then determined separately. Transmission of direct PAR was calculated depending on elevation and azimuth of the sun, the angle of incidence on roof or wall and the transmission curve of roof or wall, and the presence of non-transparent construction parts. It was assumed that neighboring greenhouse compartments fully intercept direct light and that the intensity of diffuse light coming from neighboring houses was the same as that of the unobstructed sky. Transmission of diffuse light was assumed constant according to measurements performed at an overcast sky.

Absorption of diffuse and direct light was calculated for each point independently. Absorption of a single (direct) light beam was calculated from path length through the canopy, leaf area traversed, average projection of leaves into the direction of the beam, reflection, and scattering for shaded and sunlit leaves [28] and similar to the model for hedgerow canopies [29]. Intensity of absorption of a direct PAR beam at each point in the block was calculated by multiplying outside direct light by the transmission-value of the specific direct PAR beam (through either the cover of the phytotron, the front or back wall, or the sides). Absorption of diffuse PAR (intensity was equal for all directions) was calculated by averaging absorbed intensities of numerous single beams coming from n different angles from the whole hemisphere. Crop gross assimilation was then separately calculated from absorbed total PAR fluxes for shaded and sunlit leaves ($R_{n,a,sun}$ and $R_{n,a,sun}$, respectively) to yield shaded and sunlit photosynthesis ($P_{gc,shd}$ and $P_{gc,sun}$ respectively). Total crop photosynthesis ($P_{gc}$) was the sum of those. Calculation from net $CO_2$ gas exchange to $P_{gc}$ was done by adding daytime crop dark respiration ($R_{dc}$) to $P_{gc}$. In Experiment 3, $R_{dc}$ was determined during the first four hours after darkness measuring net $CO_2$ exchange without light at a constant temperature. Temperature influence on $R_{dc}$ was calculated with a temperature-dependent mathematical function [25].

$$P_{gc,shd} = \int_{z_1,y_1,x_1}^{z_{10},y_{20},x_{20}} \left[ P_{gl,shd} \left\{ \begin{array}{c} x_1 \ldots x_{20} \\ y_1 \ldots y_{20} \\ z_1 \ldots z_{10} \end{array} \right\} \right] = P_{gl,max} \left( 1 - \frac{-\alpha_l \left[ R_{n,a,shd} \left\{ \begin{array}{c} x_1 \ldots x_{20} \\ y_1 \ldots y_{20} \\ z_1 \ldots z_{10} \end{array} \right\} \right]}{P_{gl,max}} \right) \tag{6}$$

$$P_{gc,sun} = \int_{z_1,y_1,x_1}^{z_{10},y_{20},x_{20}} \left[ P_{gl,sun} \left\{ \begin{array}{c} x_1 \ldots x_{20} \\ y_1 \ldots y_{20} \\ z_1 \ldots z_{10} \end{array} \right\} \right] = P_{gl,max} \left( 1 - \frac{-\alpha_l \left[ R_{n,a,sun} \left\{ \begin{array}{c} x_1 \ldots x_{20} \\ y_1 \ldots y_{20} \\ z_1 \ldots z_{10} \end{array} \right\} \right]}{P_{gl,max}} \right) \tag{7}$$

Shaded leaves intercepted diffuse light and the scattered part of direct light and sunlit leaves intercepted in addition to that also the direct light beam. Leaves were assumed to have the so-called near-planophile vertical leaf angle distribution. For that, a scattering coefficient ($\sigma = 0.15$) and an extinction coefficient for diffuse radiation ($K_{dif} = 0.8$) were used [30]. The photosynthesis species-specific model parameters were parameterized for the tomato cultivar 'Pannovy' with measured light-response curves (LI-6400, LI-COR Biosciences, Lincoln, NA, USA), of single leaves in three layers in the crop at three different temperatures (Experiment 3).

## 3. Results and Discussion

### 3.1. Air Exchange Rates of Closed Cabins

The closed greenhouse cabins are almost airtight. Blower-door measurements (EN 13829, 2000) in the cabins at a pressure difference of the cabin to the outside air of 50 Pa resulted in air exchange rates of around 2 $h^{-1}$. Estimated leakage areas were according to LBL ELA @ 4 Pa of approximately 45 $cm^2$ per cabin. Due to this minimal leakage, the air exchange rates of the cabins derived from the $CO_2$ depletion curves at zero pressure difference were at a very low range of 0.01 $h^{-1}$ or 1.07 $m^3$ $h^{-1}$ (Figure 2). There is a slight increase in the air exchange rate to about 0.045 $h^{-1}$ or 4.8 $m^3$ $h^{-1}$ when the air from the cabin is blown through the troughs for heat and humidity removal (Figure 2). Over a cultivated area of 24 $m^2$, this results in 0.44 $m^3$ $m^{-2}$ $h^{-1}$. When the $CO_2$ concentration in the cabin is close to the ambient $CO_2$ concentration, these low air exchange rates can be disregarded. Even a difference between cabin and external $CO_2$ concentration of 500 μmol $mol^{-1}$ at an air exchange rate of 0.045 $h^{-1}$ would result in a $CO_2$ leakage of only ~200 mg $h^{-1}$ per $m^2$ cultivated area. For comparison, Körner et al. (2007) [13] gave an estimate in their experimental facility of an air exchange rate of approximately 1.4 $m^3$ $m^{-2}$ $h^{-1}$, a typical value for a modern conventional greenhouse [31].

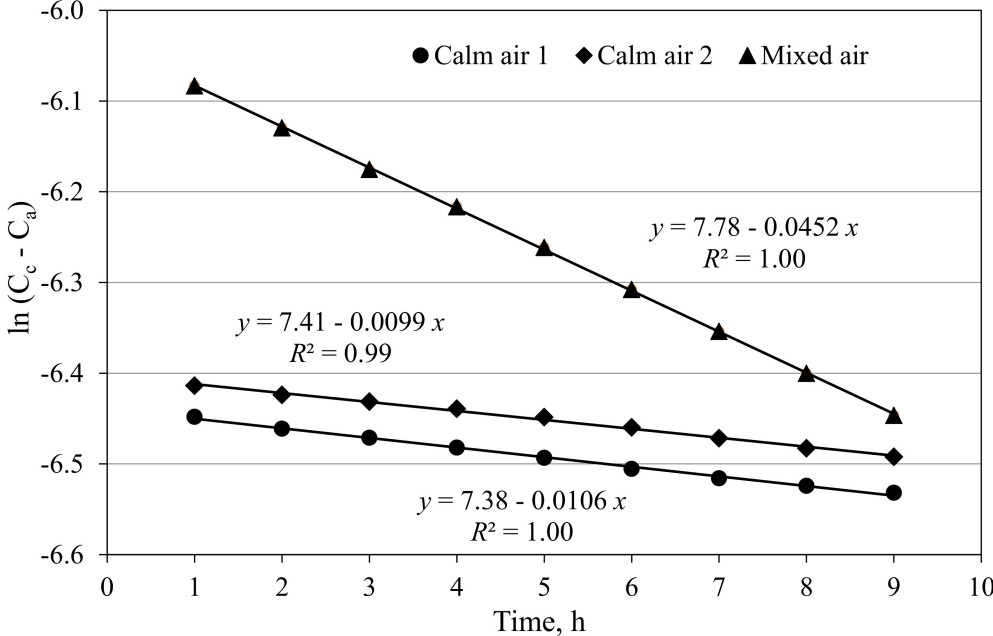

**Figure 2.** $CO_2$ depletion curves of two closed greenhouse cabins with calm air and one cabin where the air is blown through the open troughs, resulting in air exchange rates of 0.0099, 0.0106 and 0.0452 $h^{-1}$, respectively.

### 3.2. CO2 Sources and Sinks in the Cabins

$CO_2$ sources and sinks in the greenhouse cabins can be disregarded in most cases when plants are grown hydroponically. However, with small plants and low ground coverage, there may be significant effects on the measurements, which must be considered in the calculation of crop $CO_2$ gas exchange (Equation (1)). The mean daily differences between exhaust and supply air $CO_2$ concentrations at low air exchange rates (100 $m^3$ $h^{-1}$) were between 0.17 and 0.91 μmol $mol^{-1}$. This corresponds to mean daily $CO_2$ sources between 1 and 6 mg $h^{-1}$ $m^{-2}$, which can be disregarded (Figure 3). Körner et al. (2007) [13] reported an uncontrolled $CO_2$ source in their greenhouses in the range of 100 mg $m^{-2}$ $h^{-1}$ while most other authors did not present anything but disregarded such a source or sink. However, it is important to note that using irrigation water with high carbohydrate content can result in a significant root zone $CO_2$ source.

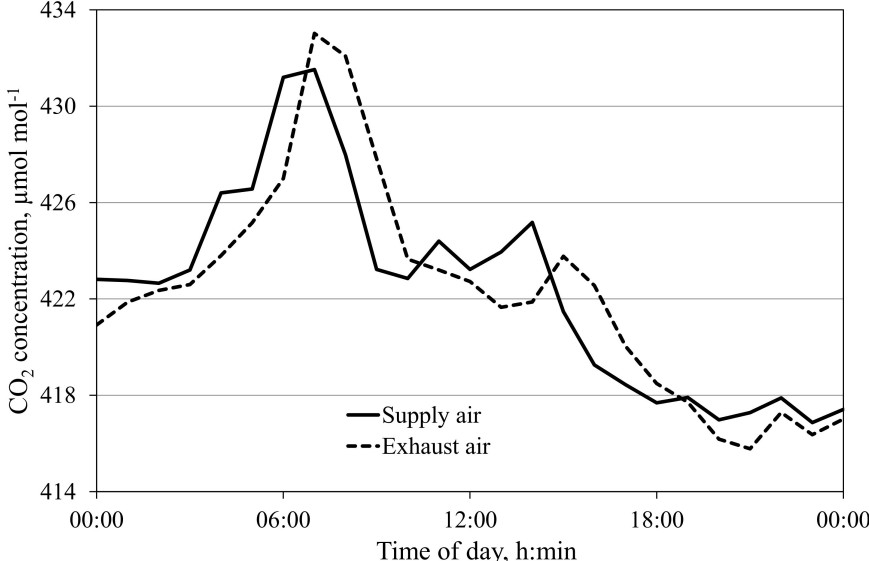

**Figure 3.** Daily time course of $CO_2$ concentration of the supply and exhaust air of an empty cabin at an air exchange rate of 3.5 $m^3$ $h^{-1}$ per $m^2$ cabin floor area.

In addition, plant cultivation in soil or substrates contains organic materials so the decomposition of the organic compounds may become a significant $CO_2$ source. As net $CO_2$ gas exchange is measured in complete crops including substrates, processes such as root respiration and the exudation of organic compounds and their decomposition by microorganisms are included. Kläring et al. (2014) [10] demonstrated that the latter may become a significant $CO_2$ source in the system and lead to an overestimation of the plant's respiration. Therefore, the possibility to separate the root from the shoot zone and measure the gas exchange in both environments separately was a logical consequence.

At low air exchange rates of ~1 $h^{-1}$, a short delay in measured $CO_2$ concentration response between exhaust air and supply air is expected (Figure 3). In this figure, the maximum absolute difference between supply and exhaust air $CO_2$ concentration is below 5 µmol $mol^{-1}$, a potential error in the gas exchange calculation for the corresponding time point of 35 mg $CO_2$ $m^{-2}$ $h^{-1}$. While this value is still relatively small (and can be disregarded in most cases), possible significant errors at high variations in external $CO_2$ concentration and low supply air rates are possible. In such cases, the scattering effect should be taken into account, e.g., by measuring or calculating the $CO_2$ concentration for any empty cabin $C_0$ and using $C_0$ instead of $C_s$ in Equation (1).

### 3.3. Possible Ranges of Environmental Conditions during the Measurements

One general problem with gas exchange measurements in greenhouses is the rising temperature in the cabin with increasing outside temperature and radiation. The facility was designed without an exterior shading screen that could be closed during high global solar radiation and thus could reduce heat penetration into the greenhouse. Without these screens, vents need to open to avoid overheating at certain conditions with high energy load. In Figure 4, outside global solar radiation just before the first daytime vent opening and after last daytime closing is illustrated from 26 March to 16 May 2016 (Experiment 2). In that period, the tomato crop occupied 24 $m^2$ and the leaf area index (LAI) increased from 1.4 to 2.2 $m^2$ $m^{-2}$. The supply air rate was 200 $m^3$ $h^{-1}$. Surprisingly, even at very low differences between cabin air temperature and ambient air temperature, the vents could be kept closed in the morning during relatively high global solar radiation ($Q_b$). In the afternoon, however, the vents closed almost as expected when global radiation fell to values close to the thresholds calculated above (Figure 4). Note that the latent heat transfer here, $q_{latent}$, is a rough estimate based on an assumed transpiration of 120 g $m^{-2}$ $h^{-1}$, a good value for a tomato crop. In other cases, the exact numbers depend on additional variables, such as LAI, the ratio of cultivation area to cabin floor area, global

radiation, and the water vapor pressure deficit (VPD, kPa) of the air. The latter can be controlled in certain constraints by controlling the supply air rate. In the present example, at a constant supply air rate, VPD was significantly lower during high global solar radiation before the vents opened than during low global radiation after the vents closed, resulting in a similar transpiration rate in both situations.

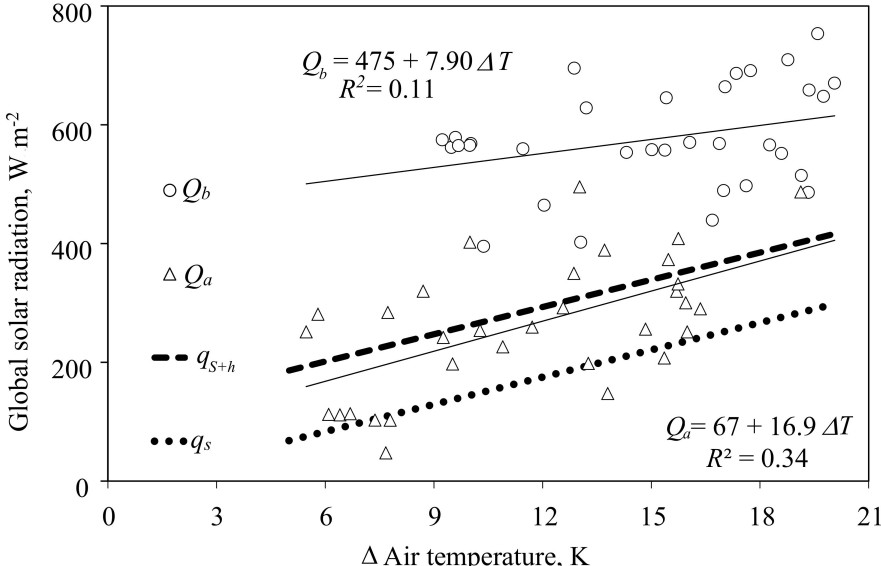

**Figure 4.** Constraints of outside global radiation and air temperature for the gas exchange measurement of an adult tomato crop in the period from 22 March to 16 May 2016 at supply air rates of 200 $m^3$ $h^{-1}$. A total of 33 data points depict the hourly average of the outside global radiation (W $m^{-2}$) as a function of the difference of the air temperature in the cabin and outside the greenhouse ($\Delta T$, K) during the hour before ($Q_b$) or after ($Q_a$), when the roof vents opened for the first or last time of the day, respectively. On the remaining 23 days, the vents did not open at any time. Solid lines represent regression lines and dotted lines represent the calculated constraints of the global radiation according to the sensitive heat transfer, $q_s$ (W $m^{-2}$), and sensitive plus latent heat transfer $q_{s+l}$ (W $m^{-2}$) out of the cabin. Transpiration of 120 g $m^{-2}$ $h^{-1}$ related to a cultivation area of 24 $m^2$ and a transmission coefficient of 0.6 of the cabin cover for global radiation were assumed.

Consolidated, during morning hours the vents are kept closed longer than expected from calculations. Thus, gas-exchange measurements could be continued during these hours. There are obviously elements in the cabins with significant heat capacity, which act as cold storage during daytime, removing heat energy from the air in the morning hours, and heating the air as cabin air temperature drops during the night. One important heat sink is likely to be the concrete floor, which covers 60% of the cabin floor. Figure 5 depicts the temperature courses of the floor, the cabin and ambient air, and the global radiation on a sunny day in May. Interestingly, between 10:00 h and 19:00 h, when the outside air temperature was high, the differences in temperature between the cabin air and the concrete floor was on average 6 K. This is higher than the temperature difference between cabin and ambient air.

There are also research topics where air exchange with the environment is undesirable. In particular, when gaseous air pollutants are supplied in order to investigate their effect on photosynthesis and plant growth, any loss of supply gases must be minimized.

Figure 6 illustrates two cooling options: In one cabin plants were set to a level 30 cm above the cabin floor and the trough cooling system was used to remove heat and water vapor from the cabin air (Figure 1); in another cabin, the air temperature was controlled by increasing the supply air rate with increasing temperature. Both approaches to removing heat from the cabin air were similarly effective (Figure 6).

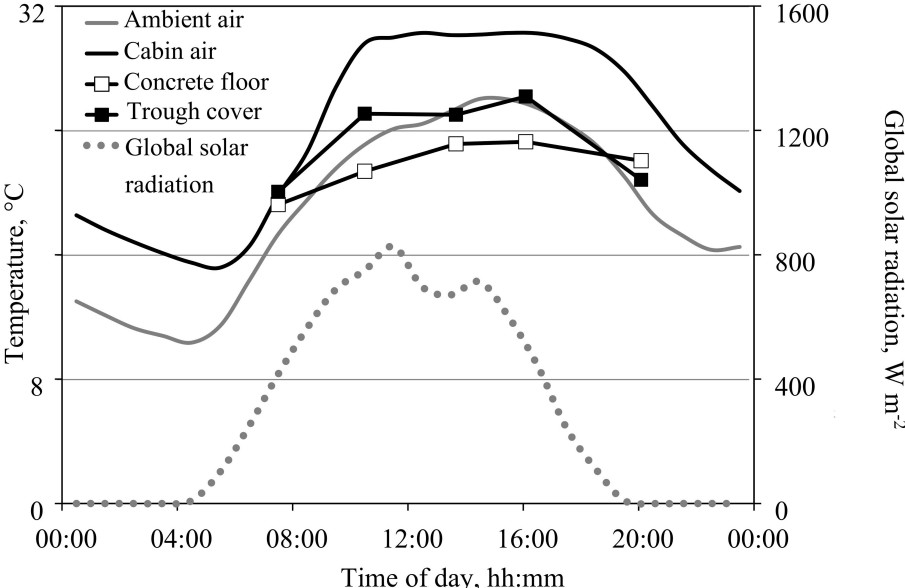

**Figure 5.** Daily courses on 11 May 2016 of global radiation outside the greenhouse and temperatures of the cabin and ambient air, the concrete cabin floor, and the chipboard covers of the root troughs. The latter two were measurements from six points in two rows each using a handheld infrared thermometer, while the others were records from the greenhouse computer. The setpoints for heating the greenhouse air during the night and the day and ventilating the greenhouse were 15, 19, and 29 °C, respectively. The temperature in the root troughs was adjusted at 19 °C. The supply air rate was 60 and 200 m³ h⁻¹ during the night and day, respectively. In addition, roof ventilation was open from 10:00 to 17:00. The same pattern of the cabin floor temperatures was obtained when the cabin was cooled by increasing the supply air rate with increasing cabin temperature and roof ventilation remained closed.

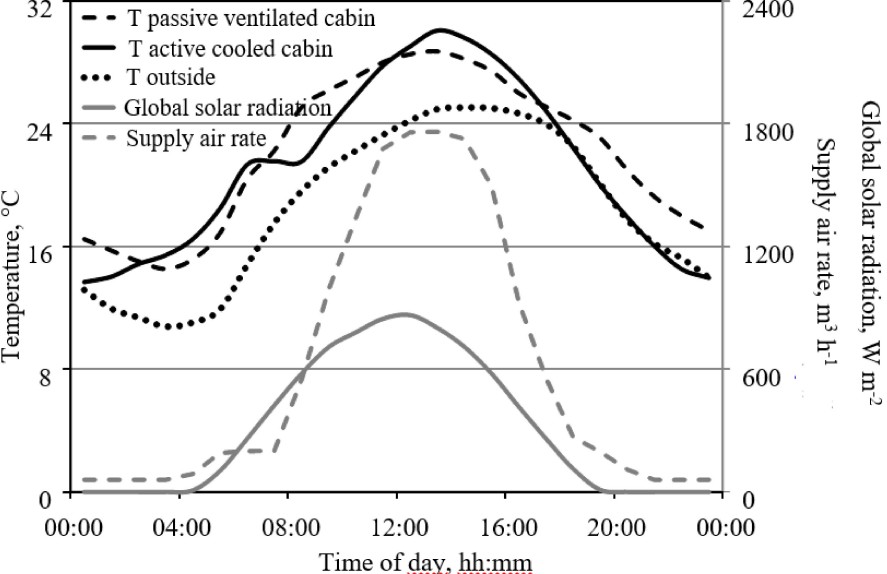

**Figure 6.** Daily course of temperature in two cabins on 10 May 2016. One cabin was cooled by increasing the supply air rate with increasing cabin temperature, while the other was cooled by blowing cabin air into the cooled troughs. The maximum supply air rate corresponds to an exchange rate of the cabin air of about 16 h⁻¹. In both cabins, a fully developed tomato crop was cultivated. Set points for heating at night and day and opening the greenhouse roof windows were 15, 19, and 32 °C, respectively.

### 3.4. Measurement of Photosynthesis in the Closed Chamber Mode

Figure 7 illustrates an example of a measured PPFD-net photosynthesis response curve of a tomato crop in a closed chamber mode. One drawback of this mode is the discontinuous supply rate of $CO_2$ to the cabin when multiple cabins are used, as there is a time lag between measurements and the $CO_2$ supply. This leads to dynamics measured photosynthesis, especially under changing outside environmental conditions. As a result, $R^2$ decreased from 0.96 to 0.91 when hourly instead of smoothed values were used (Figure 6). Therefore, this approach is not applicable if a very high temporal resolution is required. In addition, further efforts are required to improve the $CO_2$ measurement cycle and supply of $CO_2$ to the cabins.

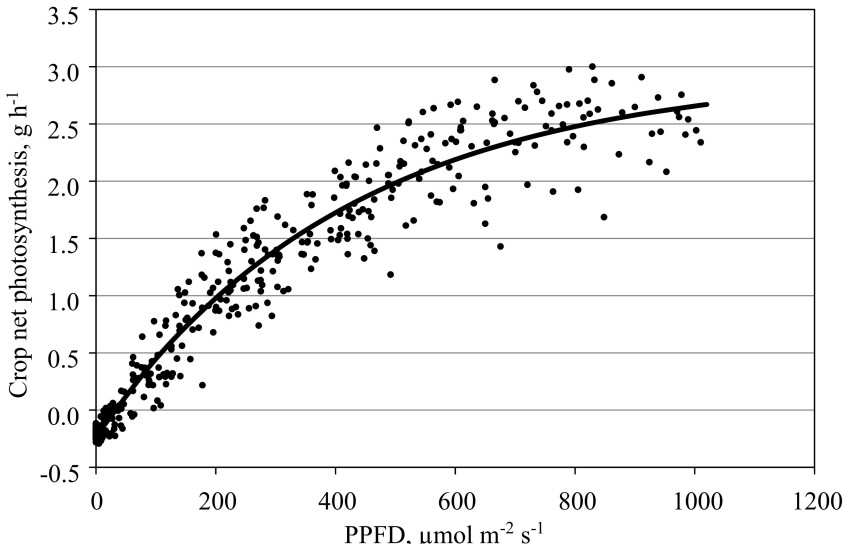

**Figure 7.** Net photosynthesis of a tomato crop measured in the closed chamber mode affected by PPFD. Data shows hourly averages of measurements smoothed over three hours in one cabin in the period from 01 April to 02 May 2016. Leaf area index increased during this period from 1.4 to 2.0 $m^2$ $m^{-2}$. $CO_2$ concentration ranged between 500 μmol $mol^{-1}$ at night and 330 μmol $mol^{-1}$ during the day. The line depicts the negative exponential PPFD response curve fitted with the data.

A few defective measurements of the net photosynthesis (technicians in the chamber training plants, harvesting fruit or maintaining sensors, open roof ventilation in March) were replaced by values estimated using the negative exponential PPFD response curve (Figure 7). Dry matter obtained from harvested fruit, removed leaves and harvested complete plants by the end of the growing period was 610 g $m^{-2}$, while the initial dry matter at planting was 3 g $m^{-2}$. Assuming a carbon content in the dry matter of 0.4 g $g^{-1}$ results in 243 g $m^{-2}$ carbon in the biomass produced by the plants. The total sum measured of the gas-exchange system was 250 g $m^{-2}$ carbon, thus a difference of only 2.8%.

### 3.5. Measurement of Photosynthesis in the Open Chamber Mode

Measuring photosynthesis in the open chamber mode is the standard procedure in this facility. It allows the separation of the aboveground and belowground gas exchange. In addition, under most conditions, transpiration can be measured in parallel. Controlling the cabin temperature by controlling the supply air rate prevents unwanted roof ventilation as long as the outside temperature is low enough (Figure 6). Figure 8 shows data of the net photosynthesis PPFD response for a tomato crop measured in two cabins. Data is in good agreement with results obtained by other scholars [15,19]. Despite the biological variation of the plants, the measurements in both cabins yielded similar results, an important requirement when comparing treatments in different cabins.

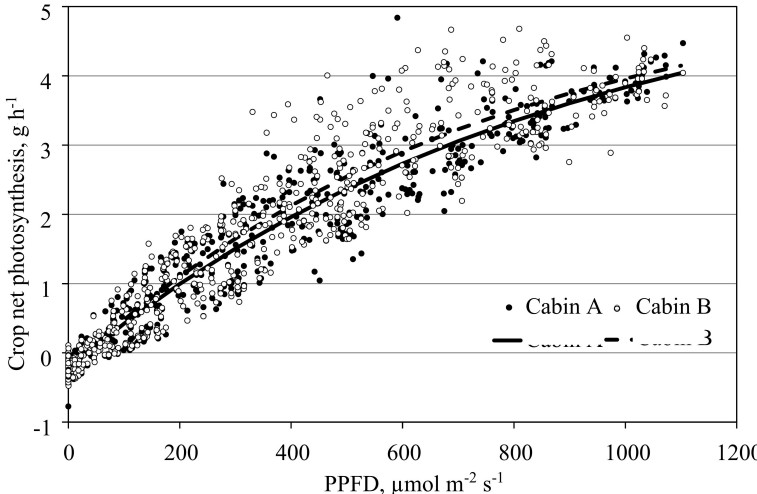

**Figure 8.** Net photosynthesis of a tomato crop measured in the open chamber mode affected by photosynthetic photon flux density (PPFD). Data shows hourly averages of measurements in two cabins in the period from 1 April to 16 May 2016. Leaf area index increased during this period from 1.5 to 2.2 $m^2$ $m^{-2}$. $CO_2$ concentration ranged between 500 μmol $mol^{-1}$ at night and 350 μmol $mol^{-1}$ during the day. The continuous and dotted lines depict the negative exponential PPFD response curves fitted with the data of the two cabins separately.

Due to a faster $CO_2$ distribution of the injected air in open chamber mode, the dynamics of measured photosynthesis due to delays of $CO_2$ supply is much smaller compared to the closed chamber mode. The best temporal resolution can be obtained when assimilated $CO_2$ is replaced by the injected air only (Figure 9). The shortest temporal resolution depends on the number of cabins in the measurement sequence. If all cabins and the outside conditions are included (standard) the complete cycle duration is 9 min. Measuring photosynthesis without supplying $CO_2$, however, restricts the upper range of $CO_2$ concentration in the chamber to slightly below that of outside $CO_2$ concentrations.

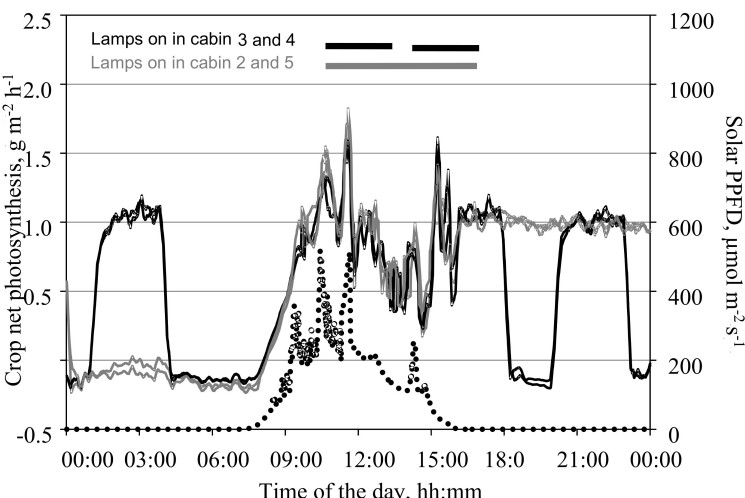

**Figure 9.** Daily course of net photosynthesis of a tomato crop measured in the open chamber mode and PPFD on the top of the crop on 24 November 2015 under two different light regimens. Data shows measurements taken in 9-min intervals. Leaf area index was 1.4 $m^2$ $m^{-2}$. $CO_2$ concentration in the cabin air ranged between 500 μmol $mol^{-1}$ during the dark and 350 μmol $mol^{-1}$ during the light phases. In two cabins, artificial light of 210 μmol $m^{-2}$ $s^{-1}$ PPFD was supplied from 15:00 to 24:00, while plants in two other cabins received artificial light of the same intensity from 15:00 to 18:00, 20:00 to 23:00 and 01:00 to 04:00.

### 3.6. Measurement of Transpiration in the Open Chamber Mode

In Figure 10 transpiration in open chamber mode in two cabins is illustrated. In this case, evaporation from surfaces can be disregarded as the root environment was covered (chipboard) and the air in the root zone was close to saturation. Increasing variation in transpiration occurred as global radiation increased (Figure 10a). Most data points were obtained at supply air rates of 60 m³ h⁻¹ at night and 200 m³ h⁻¹ during the day. Under such conditions, transpiration did not exceed a rate of 120 g m⁻² h⁻¹. However, in order to cool and dehumidify the cabin air, the supply air rate was on several days increased to a maximum of 1800 m³ h⁻¹ (Figure 6). Dehumidification was accompanied by a remarkable increase in VPD and a lower VPD gradient close to the leaf surface. This resulted in a strong increase in transpiration (Figure 10b). Interestingly, no effect of the supply air rate on photosynthesis was observed (Figure 7). Similar to the crop $CO_2$ exchange measurements, the level of transpiration for both cabins was almost equal. Figure 11 depicts the time-related resolution of the transpiration measurements under rapidly changing radiation conditions. Monitoring $H_2O$ gas exchange directly allows for a much higher time resolution of the transpiration compared to the traditional methods, i.e., lysimeters on intact plants [32].

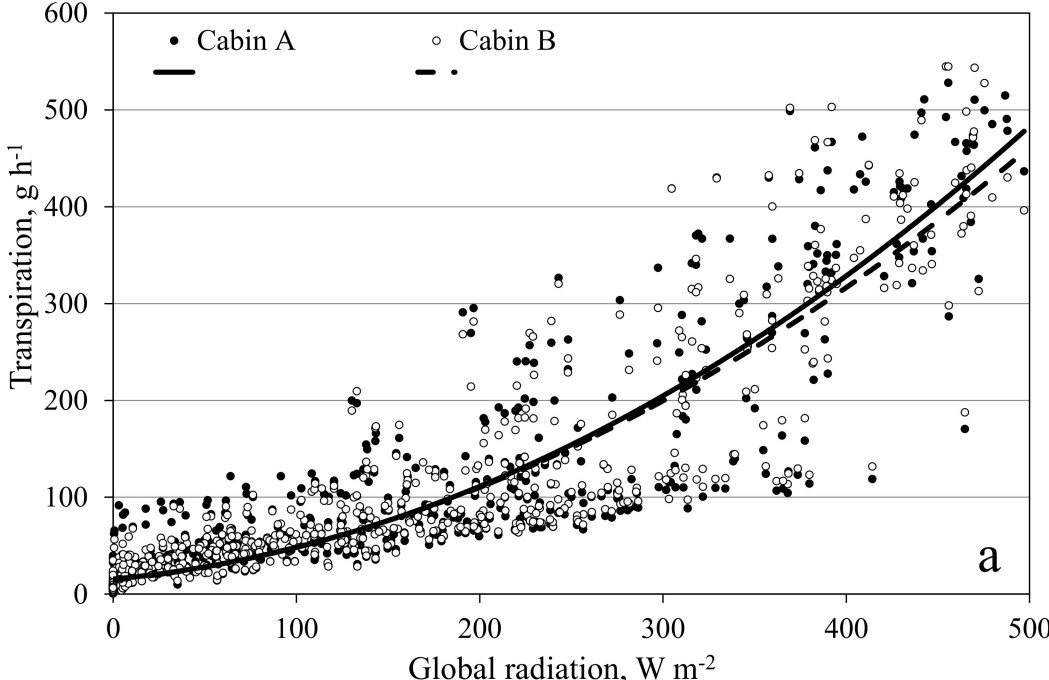

**Figure 10.** *Cont.*

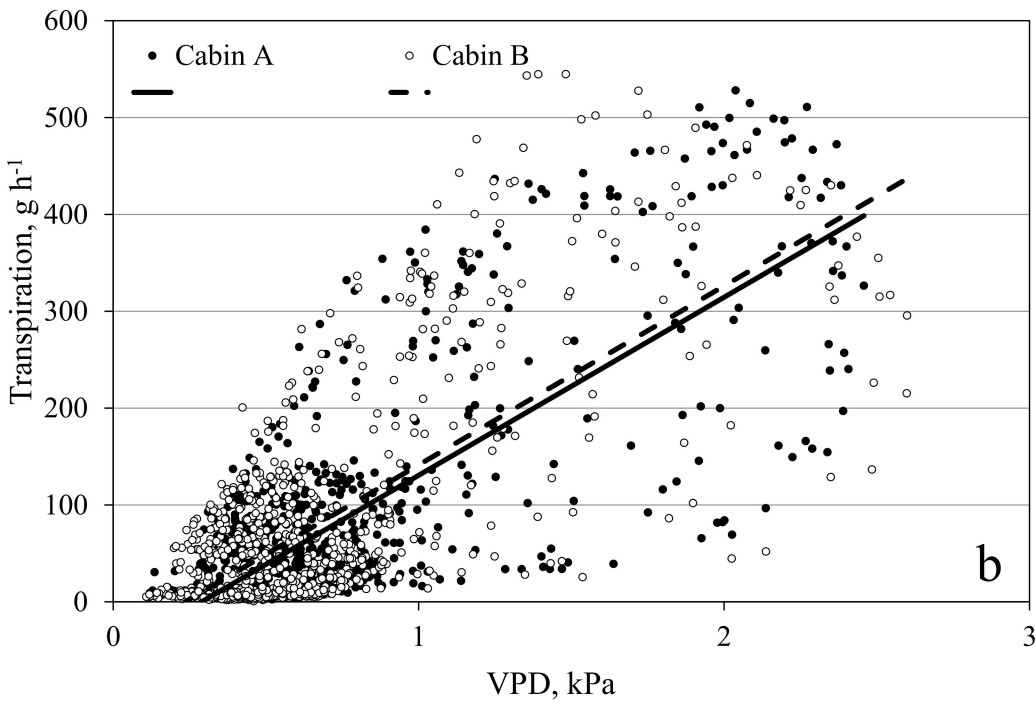

**Figure 10.** Transpiration of a tomato crop measured in the open chamber mode affected by global radiation (**a**) and vapor pressure deficit (VPD) (**b**). Data shows hourly averages of measurements in the period from 1 April to 16 May 2016. Leaf area index increased during this period from 1.5 to 2.2 $m^2$ $m^{-2}$. Air temperature ranged between 15 °C at night and 30 °C in the middle of sunny days. The continuous and dotted lines depict the empirical quadratic (**a**) and linear (**b**) response curves fitted with the data of the two cabins separately.

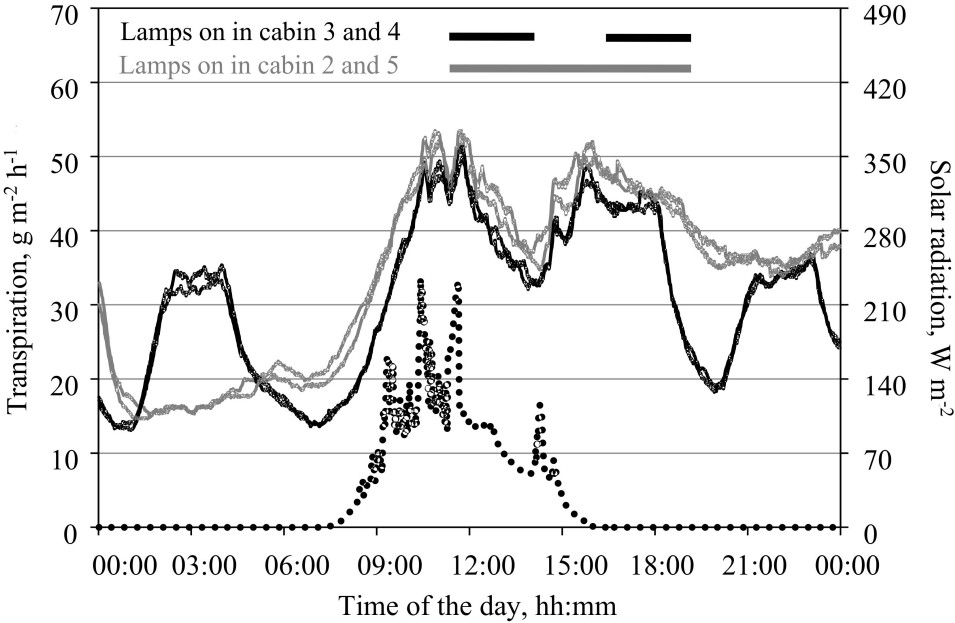

**Figure 11.** Daily course of transpiration of a tomato crop measured in the open chamber mode and global solar radiation on the top of the crop on 24 November 2015 under two different light regimens. Data shows measurements taken in 9-min intervals. Leaf area index was 1.4 $m^2$ $m^{-2}$. $H_2O$ concentration in the cabin air ranged between 8 mmol $mol^{-1}$ during the dark and 14 mmol $mol^{-1}$ during the light phase. In two cabins, lamps emitted radiation of 78 W $m^{-2}$ from 15:00 to 24:00 while plants in two other cabins received radiation of the same intensity from 15:00 to 18:00, 20:00 to 23:00, and 01:00 to 04:00.

In contrast to photosynthesis, there are no observations of precise changes in transpiration as a response to the rapid changes in radiation intensity. One explanation is the time delay of transpiration to changes of environmental conditions [33]. In addition, interaction between crop transpiration and VPD may have caused the delay in the response of the transpiration to the variation in global radiation.

### 3.7. Separating the Gas Exchange of the Shoot and of the Root Zone

Spatial separation of crop shoot and root environment enables the measurement of crop gas exchange of the shoot and root zone independently (Figure 12). To our best knowledge, such systems for greenhouses have not yet been reported in the literature. Kläring et al. (2014) [10] described such an approach for a plant cuvette that contained one fruit-bearing cucumber plant. The order of magnitude of their data is comparable with that of the present tomato experiment both for the shoot and root environment. In particular, the coupling of shoot assimilation and soil respiration in beech saplings in six connected shoot/root cuvettes positioned in a growth chamber was investigated [34]. These scholars studied the saplings' gas fluxes and carbon isotope discrimination. In contrast to the systems described in the literature [10,11], the present facility offers the possibility to control the environmental conditions of the shoot and the root environment independently, and concurrently for any cabin and trough.

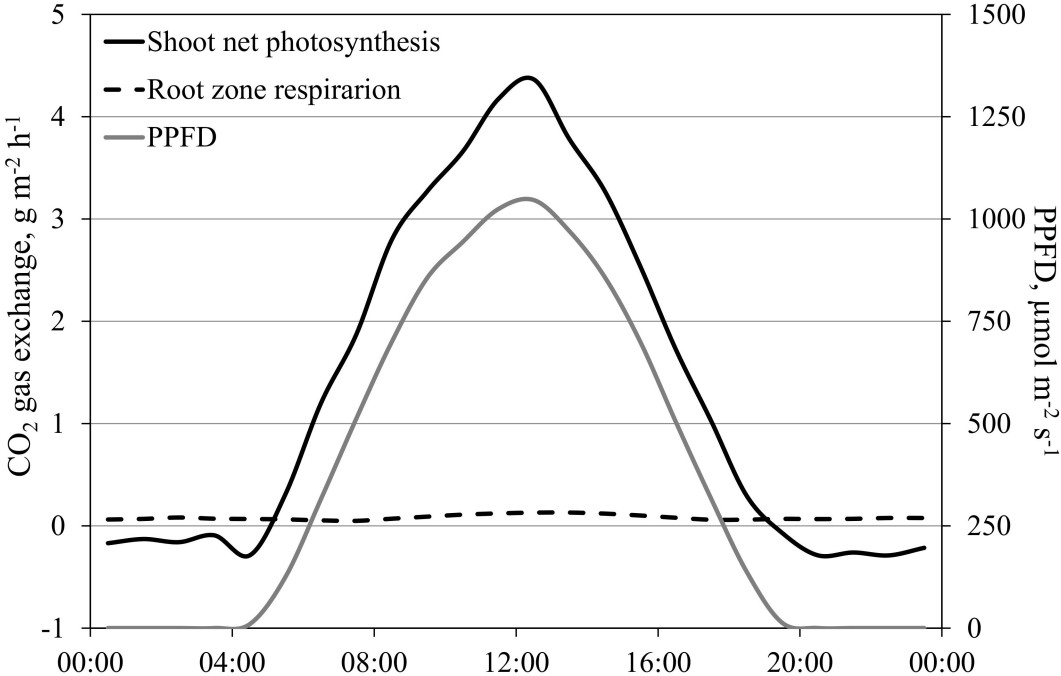

**Figure 12.** $CO_2$ gas exchange in the shoot and root zone of a tomato crop on 10 May 2016. Shoot (stem, leaves and fruit) and root dry matter were 615 and 104 g m$^{-2}$, respectively, and leaf area was 2.2 m$^2$ m$^{-2}$. Temperature in the root environment was kept constant at 19 °C, while cabin temperature rose from 15 °C at night to 30 °C in the middle of the day.

### 3.8. Models and Measurements

For the use in crop model validation, the block-shaped 3-dimensional photosynthesis model was used. For that, the tomato cultivar 'Pannovy' was parameterized and fitted to the photosynthesis parameters. The resulting model was used to compare the pure measured photosynthesis (i.e., $CO_2$ gas exchange) as discussed above with simulated photosynthesis. The results show a good fit between measured and simulated crop gross photosynthesis (Figure 13). The overall agreement as calculated with adjusted $R^2$ is with 0.976 very high. However, larger differences and underestimations of the modeled $CO_2$ gas exchange exist. Especially a difference between morning and afternoon fits are

evident (Figure 13c), which is probably the lack of a mid-day repression or hysteresis in the model. These discrepancies have been observed in earlier measurements of the same system, while however no model comparison was made [35]. However, the combination of physical gas-exchange measurements in a crop stand in combination with parameterized photosynthesis models is a powerful tool for improvement of crop photosynthesis predictions. The next steps are among others screening and parameterization of different cultivars and crops.

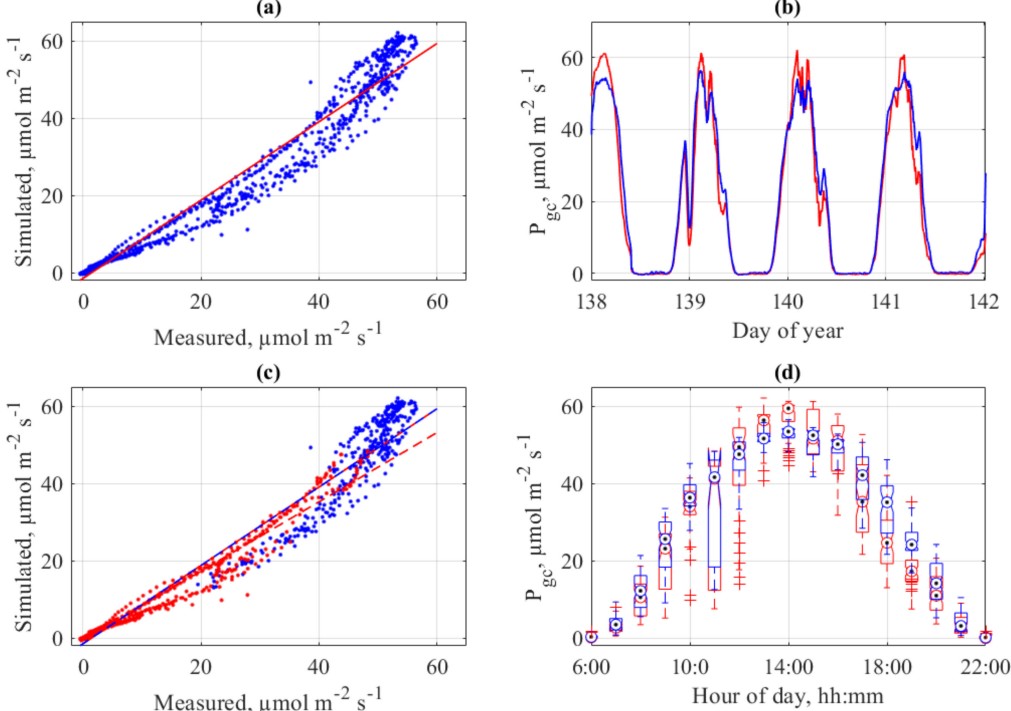

**Figure 13.** Measured and Simulated crop gross photosynthesis ($P_{gc}$) of tomato cv 'Pannovy' for four selected days in Experiment 3 with a)simulated vs. measured values (**a**; $R^2$ 0.967), time course of measured (red) and simulated (blue) (**b**), simulated vs. measured values before 12 h (red) and after 12 h (blue) (**c**), and crop gross photosynthesis ($P_{gc}$) as function of hour per day for measured (red) and simulated (blue) (**d**).

## 4. Conclusions

A multi-cabin greenhouse for measuring the gas exchange of crops was designed and constructed. $CO_2$ and $H_2O$ gas exchange of complete stands of plants can be measured in parallel in up to eight greenhouse cabins, each of which can be accessed and controlled independently. This allows for continuous monitoring of the effects of different treatments on photosynthesis, dark respiration, and transpiration without any time delay. Except for some conditions with high ambient temperatures in combination with high global radiation in summer, gas exchange measurements are possible all year round. In addition to the typical climate control options and the possibility to manipulate the conditions in the root environment, this facility provides, in particular, the possibility of introducing air pollutants and other specific plant-relevant gases into the cabins. In addition to the aboveground components, the temperature can be controlled and the gas release from the root zone can be measured independently in four troughs per cabin. This final aspect opens up new options for studying the interactions of the canopies' root and shoot metabolisms as well as a powerful tool for crop growth modeling improvement.

**Author Contributions:** Conceptualization, H.-P.K. and O.K.; methodology, H.-P.K.; software, O.K.; validation, H.-P.K., O.K.; formal analysis, H.-P.K., O.K.; investigation, H.-P.K., O.K.; resources, H.-P.K., O.K.; data curation, H.-P.K., O.K.; writing—original draft preparation, H.-P.K.; writing—review and editing, H.-P.K., O.K.; visualization,

H.-P.K., O.K.; supervision, H.-P.K., O.K.; project administration, H.-P.K.; funding acquisition, H.-P.K. All authors have read and agreed to the published version of the manuscript.

**Funding:** This research was funded by the European Regional Development Fund.

**Acknowledgments:** This construction of the research facility was supported by the European Regional Development Fund. We thank Angela Schmidt, Jörg Bigus, Ingo Hauschild, and Thomas Runge for their excellent technical support.

**Conflicts of Interest:** The authors declare no conflict of interest.

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
