# Peer review of "Design of a Real-Time Gas-Exchange Measurement System for Crop Stands in Environmental Scenarios"

_agronomy, doi:10.3390/agronomy10050737_

Round 1

Reviewer 1 Report

The title concerning the design of a real-time gas-exchange measurement system for greenhouse crop stands is appropriate and indicates the main message of the paper.  No changes necessary.

The abstract provides a concise but complete summary about the main objective, general picture of the methodological approach, results and conclusions of the paper.

Keywords are well suited with the paper text.

The introduction provides sufficient background information for readers. The general topic, issue, or area of concern is given to illustrate the context.

The material and method section enables motivated researches to repeat the demonstration.

The works is original and contain new results that significantly advance the research field.

The work is understandable, correct, and appropriate for the journal.

The results are interesting and important to researchers in plant physiology and GHG emissions.

The paper is well written, the English understandable, the length and format are appropriate.

References are appropriate and adequate to related works and covered sufficiently in the list.

The conclusions are logically supported by the obtained results.

The figures and tables are excellent, easily readable, correct and informative.

Resize table 1 for do not split the words letters (IF in the original table all words are not full) and try to move the table on a single page.

Line 512: “hwever” =    “however” 

Figure 4: “conrete”  =  “concrete”

Figure 11. “respirarion”  correction: “respiration”

Author Response

We thank Reviewer 1 for his comments and his work to review this manuscript!

There were only few critical changes asked for. We have worked on them out as:

R1: Resize table 1 for do not split the words letters (IF in the original table all words are not full) and try to move the table on a single page.

AUTHORS: We have reduced the content of Table 1 such that is now fiting on one page and all typesetting is clear

R1: Line 512: “hwever” =    “however” 

AUTHORS:Corrected

R1: Figure 4: “conrete”  =  “concrete”

AUTHORS: Corrected

R1: Figure 11. “respirarion”  correction: “respiration”

AUTHORS: Corrected

Reviewer 2 Report

I understand the authors’ motivation of designing facilities to measure the photosynthesis of complete crops. The manuscript well describes performance of the greenhouse system, including measurement of root respiration separately from shoots gas exchange, which I think is one of the advantages of the designed system. I would like to add the following comments for minor revision.

1) Introduction. Micrometeorological techniques such as eddy covariance are very common nowadays to measure gas exchange on a canopy scale and to provide useful data for crop physiological models. I recommend that the authors refer to those techniques briefly with its advantages/disadvantages when comparing with the whole crop gas-exchange greenhouse measurement systems.

2) Section 2.1 & 2.2. A figure of the cabins would be of great help for readers to understand the structure and configureuration of the experimental system.

3)L195. “the vaporisation enthalpy at 100 degree C”.

4) L197. The term “global solar radiation” is preferable to “global radiation” to avoid confusion, at least at the first appearance of the manuscript.

5) L261. Please add the full spelling of the abbreviated term, PPFD, at the first appearance.

6) L338. “per m2 of floor area”.

7) Figure 5. Explanation of “T ventilated cabin” and “T cooled cabin” is confusing.

8) Figure 6. Nighttime CO2 concentration is useless. It is better to show the range of CO2 concentration in the daytime (i.e. when PPFD is >0). This is also the case for Figure 7.

9) L438. It is better to use the unit “g CO2 m-2”, “g C m-2, or “g m-2 carbon” like in L444.

10) Figure 8. It is better to use the unit of hour instead of minute on the abscissa.

11) Figure 12. How did authors measure “gross” crop photosynthesis? There seems no description in the manuscript how to measure or to estimate above-ground crop respiration rate in the daytime.

Author Response

We thank Reviewer 2 for his very valuable comments and his work reviewing our manuscript.

We have addressed all comments and changed the manuscript accordingly. 

I understand the authors’ motivation of designing facilities to measure the photosynthesis of complete crops. The manuscript well describes performance of the greenhouse system, including measurement of root respiration separately from shoots gas exchange, which I think is one of the advantages of the designed system. I would like to add the following comments for minor revision.

1) Introduction. Micrometeorological techniques such as eddy covariance are very common nowadays to measure gas exchange on a canopy scale and to provide useful data for crop physiological models. I recommend that the authors refer to those techniques briefly with its advantages/disadvantages when comparing with the whole crop gas-exchange greenhouse measurement systems.

We thank the reviewer for this valuable comment. We have added/adjusted a section (Line 35-45)

We have added the following in the Introduction:

The interest in upscaling physiological processes in space and time has been grown since the 1980th and new measurements techniques such as eddy-covariance or remote sensing procedures have been developed [2]. While these techniques are commonly used to quantify CO2 exchange on ecosystem scale [3], there disadvantage is the large ecosystem scale and relatively long time frame of 24h. In protected cultivation in controlled environments, e.g. in greenhouses, processes with long time frame within days are next to fast time responses within seconds or minutes of high importance for optimised climate control strategies [4, 5].

Followed by adjustment of this text:

To create data for model calibration and validation on greenhouse scale and in controlled environments, for optimised climate control, and for model-based decision support systems; larger systems measuring single plants [6] or a group of small plants on laboratory scale [7] or potted plant trays [8, 9] have therefore been developed.

2) Section 2.1 & 2.2. A figure of the cabins would be of great help for readers to understand the structure and configureuration of the experimental system.

We also thank the reviewer for this comment. We have included a new figure (Figure 1):

Figure 1. Simplified sketch of the experimental gas-exchange facility (not on scale) of one of eight cabins. Red lines denote gas-supply, blue lines denote gas sampling, and green lines is the outlet air. For supplementary lighting 12 HPSL or multi-channel LED lamps can be mounted (six lamps of each side of each greenhouse compartment. All fluxes are measured with continuous flux-meters. Irrigation and fertigation is done automatically and can be adjusted for each trough independently (not in the figure).    

This also means that all other figures were renumbered.

3)L195. “the vaporisation enthalpy at 100 degree C”.

This is correct and we added ‘at 100 degree C’

4) L197. The term “global solar radiation” is preferable to “global radiation” to avoid confusion, at least at the first appearance of the manuscript.

We agree with the reviewer and we have actually changed it such that we now use the term “global solar radiation” throughout the entire manuscript, including the figures

5) L261. Please add the full spelling of the abbreviated term, PPFD, at the first appearance.

In line 274-275 we have now added: ‘photosynthetic photon flux density (PPFD, µmol [photons] m-2 s-1)’

7) Figure 5. Explanation of “T ventilated cabin” and “T cooled cabin” is confusing.

We have adjusted Figure 5 (now Figure 6) with the terms:

“T passive ventilated cabin”

“T active cooled cabin”

8) Figure 6. Nighttime CO2 concentration is useless. It is better to show the range of CO2 concentration in the daytime (i.e. when PPFD is >0). This is also the case for Figure 7.

The data shown are all data from April 01 to May 02, which includes daytime and nighttime. Nighttime measurements were used to estimate dark respiration. For a complete understanding, we have added in the Caption of Figure 6 (now Figure 7) both daytime and nighttime CO2 concentration, as all CO2 concentration data is input to the calculations of Crop net photosynthesis of Eqn. 1

9) L438. It is better to use the unit “g CO2 m-2”, “g C m-2, or “g m-2 carbon” like in L444.

We agree with the reviewer and we have corrected the language of this paragraph and re-arranged without the need the unit in L438 (is left out) to:

A few defective measurements of the net photosynthesis (technicians in the chamber training plants, harvesting fruit or maintaining sensors, open roof ventilation in March) were replaced by values estimated using the negative exponential PPFD response curve (Figure 7). Dry matter obtained from harvested fruit, removed leaves and harvested complete plants by the end of the growing period was 610 g m-2, while the initial dry matter at planting was 3 g m-2. Assuming a carbon content in the dry matter of 0.4 g g-1 results in 243 g m-2 carbon in the biomass produced by the plants. The total sum measured of the gas-exchange system was 250 g m-2 carbon, thus a difference of only 2.8%.

10) Figure 8. It is better to use the unit of hour instead of minute on the abscissa.

We have adjusted the unit to hour on all Figures with the same pattern: Figures 9, 11 and 12 (earlier numbering Figures 8, 10 and 11) 

11) Figure 12. How did authors measure “gross” crop photosynthesis? There seems no description in the manuscript how to measure or to estimate above-ground crop respiration rate in the daytime.

We understand this remark and thank the reviewer for this! We have included the following paragraph in section 2.10; L332-335

Calculation from net CO2 gas exchange to Pgc was done by adding daytime crop dark respiration (Rdc) to Pgc. In Exp. 3, Rdc was determined during the first four hours after darkness measuring net CO2 exchange without light at constant temperature. Temperature influence on Rdc was calculated with a temperature dependent mathematical function [24].